# New, Eco-Friendly Method for Synthesis of 3-Chlorophenyl and 1,1′-Biphenyl Piperazinylhexyl Trazodone Analogues with Dual 5-HT_1A_/5-HT_7_ Affinity and Its Antidepressant-like Activity

**DOI:** 10.3390/molecules27217270

**Published:** 2022-10-26

**Authors:** Przemysław Zaręba, Anna Partyka, Gniewomir Latacz, Grzegorz Satała, Paweł Zajdel, Jolanta Jaśkowska

**Affiliations:** 1Faculty of Chemical Engineering and Technology, Department of Chemical Technology and Environmental Analytics, Cracow University of Technology, 24 Warszawska Street, 31-155 Cracow, Poland; 2Department of Clinical Pharmacy, Jagiellonian University Medical College, 9 Medyczna Street, 30-688 Cracow, Poland; 3Department of Technology and Biotechnology of Drugs, Jagiellonian University Medical College, 9 Medyczna Street, 30-688 Cracow, Poland; 4Department of Medicinal Chemistry, Maj Institute of Pharmacology, Polish Academy of Sciences, 12 Smętna Street, 31-343 Kraków, Poland; 5Department of Organic Chemistry, Jagiellonian University Medical College, 9 Medyczna Street, 30-688 Cracow, Poland; 6Faculty of Chemical Engineering and Technology, Department of Organic Chemistry and Technology, Cracow University of Technology, 24 Warszawska Street, 31-155 Cracow, Poland

**Keywords:** microwave, reductive alkylation, antidepressant, serotonin, arylpiperazine

## Abstract

Serotonin 5-HT_1A_ and 5-HT_7_ receptors play an important role in the pathogenesis and pharmacotherapy of depression. Previously identified *N*-hexyl trazodone derivatives, 2-(6-(4-(3-chlorophenyl)piperazin-1-yl)hexyl)-[1,2,4]triazolo[4,3-*a*]pyridin-3(2*H*)-one hydrochloride (**7a·HCl**), with high affinity for 5-HT_1A_R and 2-(6-(4-([1,1′-biphenyl]-2-yl)piperazin-1-yl)hexyl)-[1,2,4]triazolo[4,3-*a*]pyridin-3(2*H*)-one hydrochloride (**7b·HCl**), a dual-acting 5-HT_1A_/5-HT_7_ receptor ligand, were prepared with a new microwave-assisted method. The protocol for the synthesis of **7a** and **7b** involved reductive alkylation under a mild reducing agent. We produced the final compounds with yield of 56–63% using ethanol or 51–56% in solvent-free conditions in 4 min. We then determined the 5-HT_7_R binding mode for compounds **7a** and **7b** using in silico methods and assessed the preliminary ADME and safety properties (hepatotoxicity and CYP3A4 inhibition) using in vitro methods for **7a·HCl** and **7b·HCl**. Furthermore, we evaluated antidepressant-like activity of the dual antagonist of 5-HT_1A_/5-HT_7_ receptors (**7b·HCl**) in the forced swim test (FST) in mice. The 5-HT_1A_R ligand (**7a·HCl**) with a much lower affinity for 5-HT_7_R compared to that of **7b·HCl** was tested comparatively. Both compounds showed antidepressant activity, while 5-HT_1A_/5-HT_7_ double antagonist **7b·HCl** showed a stronger and more specific response.

## 1. Introduction

In 2020, more than 280 million people worldwide suffered from depression [1]. Due to the COVID-19 pandemic, an increased number of mental disorders was observed in the years 2020–2022. A large body of evidence indicates that emotional disorders, depression, insomnia and anxiety are common among people quarantined during a pandemic [2,3]. Due to the growing percentage of patients and the limited effectiveness of current therapeutic methods, the search for new antidepressants is one of the leading research directions in pharmaceutics worldwide.

The 5-HT_1A_ receptor (5-HT_1A_R) has been proposed as a biological target for developing antidepressant drugs. The 5-HT_1A_R activates various effectors through the G_i/o_ proteins, and its stimulation causes inhibition of adenylyl cyclase [4]. It is also involved in the K^+^ and Ca^2+^ ion pathways and the stimulation of phospholipase C, as well as the activation of the mitogen-activated protein kinase Erk2 [5]. The 5-HT_1A_R is strongly expressed in neurons as a presynaptic inhibitory autoreceptor, but also as a postsynaptic heteroreceptor in many areas of the brain [6]. The C(-1019)G functional polymorphism (rs6295), which regulates the 5-HT_1A_R gene, has been identified and associated with an increased risk of affective disorders and resistance to treatment with selective serotonin reuptake inhibitors (SSRIs) [7,8].

The 5-HT_7_ receptor (5-HT_7_R) also plays an important role in the pathogenesis and pharmacotherapy of affective disorders [9,10]. It is coupled with the G_s_-protein, and its stimulation results in the activation of adenylyl cyclase. Due to its coupling with the G12 protein, it can activate Rho family GTPases (CDC42 and RhoA) [5,11]. It has been shown that the 5-HT_7_R/MMP-9 pathway is specifically activated in the hippocampus during chronic stress, which is the key for inducing depressive-like behavior [10].

In some studies, 5-HT_1A_R and 5-HT_7_R overexpression in non-neuronal cell lines has been used to evaluate their coupling to several transduction pathways [12]. The antidepressant efficacy of 5-HT_1A_R and 5-HT_7_R ligands depends as much on the response system characteristics as on the physicochemical properties of the ligands, distribution in the brain and the number of receptors expressed in the brain region [13]. Of note, the inactivation or blockade of 5-HT_7_R has behavioral effects similar to those of antidepressants [13]. The animal studies have shown the upregulation of 5-HT_7_R in the hippocampus after stress exposure [14]. The forced swim test (FST) in mice confirms the antidepressant effect of both the selective 5-HT_7_R antagonist [15] and the multifunctional 5-HT_2A_/5-HT_7_/D_2_ receptors antagonists [16]. Previously published in vivo studies indicate antidepressant activity of dual-acting 5-HT_1A_/5-HT_7_ receptor antagonists in animal models [17]. However, the compounds show higher affinity for 5-HT_1A_R than for 5-HT_7_R. The tested compounds show significantly and dose-dependently reduced immobility by 46%, detected in FST in mice [17].

Our recent interest has been focused on trazodone, a widely used antidepressant drug. It belongs to a class of long-chain arylpiperazines (LCAP) and behaves as a serotonin 5-HT_2A_R antagonist and serotonin reuptake inhibitor. When studying the influence of structural modifications in the alkylene linker on serotonin receptor affinity, we shifted the orientation of the receptor profile from dominant 5-HT_2A_R to 5-HT_1A_R upon an extension of the carbon linker in LCAP [18,19]. The effect was valid for 2-(6-(4-(3-chlorophenyl)piperazin-1-yl)hexyl)-[1,2,4]triazolo[4,3-*a*]pyridin-3(2*H*)-one hydrochloride (**7a·HCl**), a direct hexyl analogue of trazodone. Compound **7a·HCl** showed high affinity for the 5-HT_1A_R (*K*_i_ = 16 nM) and moderate affinity for the 5-HT_7_R (*K*_i_ = 278 nM) (Figure 1).

This group also comprised 2-(6-(4-([1,1′-biphenyl]-2-yl)piperazin-1-yl)hexyl)-[1,2,4]triazolo[4,3-*a*]pyridin-3(2*H*)-one hydrochloride (**7b·HCl**), dual-acting 5-HT_1A_/5-HT_7_ receptor ligand, with equipotent activity for 5-HT_1A_R (*K*_i_ = 20 nM), 5-HT_7_R (*K*_i_ = 19 nM) [18].

Target compounds **7a** and **7b** were synthesized using a two-step method involving microwave-assisted *N*-alkylation reactions. However, despite the short reaction time and the limited use of solvents and toxic reagents, the method required the use of an excess of the alkylating agent (three equivalents) in the reaction of [1,2,4]triazolo[4,3-*a*]pyridin -3(2*H*)-one with 1,6-dibromohexane [18]. Thus, we decided to optimize the synthesis, trying to obtain compounds **7a** and **7b** using reductive alkylation of amines with carbonyl compounds [20]. This approach is designed to eliminate the need to use an excess of the alkylating agent.

Next, we assessed the absorption, distribution, metabolism, excretion and toxicity (ADMET) parameters using in silico methods, followed by an assessment of the functional profile of compounds **7a·HCl** and **7b·HCl** at the 5-HT_1A_ and 5-HT_7_ receptor, safety profile of the compound **7b·HCl** in hepatoma HepG2 cell line and CYP3A4 inhibition. Then, we investigated the antidepressant-like activity of compounds **7a·HCl** and **7b·HCl** in the forced swim test (FST) in mice. In order to test whether, in the case of trazodone analogues, the dual activity of **7b·HCl** at 5-HT_1A_/5-HT_7_ receptors improved the antidepressant properties in comparison with the more selective 5-HT_1A_R ligands, we also evaluated **7a·HCl** with stronger affinity for 5-HT_1A_R.

## 2. Results and Discussion

### 2.1. Eco-Friendly Method of Synthesis

We started our work with the development of an effective synthesis method for compounds **7a** and **7****b** by reductive alkylation. In an earlier publication, we obtained products **7a** and **7b** in a two-step *N*-alkylation pathway. In step i (Figure 1), we conducted reactions between [1,2,4]triazolo[4,3-*a*]pyridin-3(2*H*)-one (**1**) and 1,6-dibromohexane (**2**, three equivalents) in the presence of microwave irradiation (MW). Then, the resulting intermediate (**4**) was reacted with arylpiperazine (**6a** or **6b**) in step iv.

In this article, we present the results of our tests on an alternative reaction pathway, via reductive alkylation. In step ii (Figure 1), we obtained 2-(6,6-dimethoxyhexyl)[1,2,4]triazolo[4,3-*a*]pyridin-3(2*H*)-one (**5**) in the microwave-assisted reaction of [1,2,4]triazolo[4,3-*a*]pyridin-3(2*H*)-one (**1**) with 6-chloro-1,1-dimethoxyhexane (**3**) (1 equivalents). Microwave reactions were carried out in an open-vessel mode (round bottom flask). Importantly, analogous conditions to the previously developed method of obtaining **4** were used at this stage; however, stoichiometric amounts of the alkylating agent (**3**) were used. The use of a threefold excess of alkylating agent **2** in the preparation of **4** [18] was necessary due to the competitive substitution reaction leading to obtaining 2,2′-(hexane-1,6-diyl)di([1,2,4]triazolo[4,3-*a*]pyridin-3(2*H*)-one. Similar cases of formation of a disubstituted product were also observed in other *N*-alkylation reactions with dihaloalkanes [21]. The use of a large excess of the alkylating agent is problematic both because of its toxicity and the difficult removal of the excess from the reaction mixture. Especially for the synthesis on a greater scale, the removal of a large excess of an unreacted alkylating agent is economically disadvantageous. In the case of synthesis of **5**, this problem did not occur due to the structure of the alkylating agent (**3**), which allowed the use of equimolar amounts corresponding to **1**. The product was obtained with a high yield (89%), and in step iii, it was hydrolyzed in a 10% HCl solution to 6-(3-oxo [1,2,4]triazolo[4,3-*a*]pyridin-2(3*H*)-yl)hexanal (**5**) with a yield of 86% after phase separation and evaporation. In step v, the resulting product (**5**) was reacted with the appropriate arylpiperazine (**6a** and **6b**) to obtain the final product (**7a** and **7****b**).

In the case of the reductive alkylation step v, we started by checking the possibility of obtaining **7a** by applying the synthesis conditions described in the literature and developed for the preparation of other LCAP [21]. This method requires a strong reducing agent NaBH(OAc)_3_ (two equivalents) in CH_2_Cl_2_. Here, steps v (imine formation) and vi (imine reduction) take place simultaneously; however, the reaction medium is strongly sensitive to moisture. Product **7a** was obtained with a 62% yield. We also carried out the reactions in the variant of the synthesis assisted by microwave radiation, obtaining a yield of 69% in 1 min. Then, we decided to develop a new, microwave-assisted method, using a milder and less moisture-sensitive reducing agent, NaBH_4_ (two equivalents). In the first tested variant, we tried to carry out steps v (imine formation) and vi (imine reduction) simultaneously, like in the literature conditions. We conducted the reactions with the simultaneous presence of substrates **5**, **6a** and the reducing agent. Conducting the reactions under conventional conditions, at room temperature or at reflux, no final amine was observed in the reaction mixture, but only the product of the aldehyde reduction to alcohol. In the MW variant, the post-reaction mixture contained about 50% of alcohol from the reduction of aldehyde **5**. Final product **7a** was isolated with a yield of 16%. A much better effect was achieved by carrying out the reductive alkylation in stages. In step v, the appropriate imine was obtained in the reactions of **5** and **6a**, carried out for one minute in a microwave reactor. Then, the reducing agent was added, and the reaction continued for another three minutes (step vi). The product was obtained with a yield of 44% in ethanol, and the rest was the products of imine hydrolysis (starting materials, **6a** and **5**) as well as the product of reduction of aldehyde **6a** to alcohol. In the solvent-free variant, the yield was reduced to 16%. Interestingly, the use of catalytic amounts of iron (II) phthalocyanine (FePC) (0.1 equivalents) led to an increase of the yield of the obtained product to 56% in ethanol (EtOH) or 51% in solvent-free synthesis. Even higher yields in this variant were obtained for product **7b** (Table 1).

Importantly, the method presented here is characterized by a reduction in the reaction time and the replacement of halogenated solvents with ethanol, compared to the conventional method of reductive alkylation. On the other hand, this method allows for a threefold reduction in the amount of the alkylating agent used when compared to the *N*-alkylation method. In the context of both previously developed reactions, the reduction of the use of hazardous reagents and toxic solvents significantly improves the ecological value of the process. The new method of synthesis is an interesting solution, especially considering larger-scale production, due to the reduction of the amount of waste generated.

### 2.2. Molecular Modeling

Interesting results of radioreceptor and functional research prompted us to analyze the interactions in protein–ligand complexes in order to select fragments in the ligand structure that are of key importance for receptor affinity. We analyzed the binding mode of **7b** in 5-HT_1A_R and **7a**, **7b** in 5-HT_7_R by docking to the crystal structure of 5-HT_1A_R in the complex with aripiprazole [22] (Figure 2A) and the homologous model of the 5-HT_7_R in an inactive state (template 5-HT_1B_R; id: 4IAQ), obtained from the GPCRdb [23,24] (Figure 2B).

In the case of 5-HT_1A_R, the obtained binding mode corresponded to aripiprazole from the crystal structure of the complex. Both **7b** and the reference compound had the ability to form a salt bridge in the piperazine ring with D3.32. Differences appeared in the orientation of the aryl rings on piperazine; however, this was due to the nature of the 2-biphenyl substituent. In 5-HT_7_R, we obtained two active **7b** conformations. In both, the triazolopyridinone moiety was folded in the same manner to form π-cation interactions with R7.36, and salt bridges were observed on the piperazine ring with D3.32. Differences appeared in the orientation of the biphenyl rings. In the first one (light blue), the first ring, attached to piperazine, was facing helices 5 and 6, creating π-stacking with F6.51 and F6.52. The second ring bent towards helix 3, creating π-stacking with F6.52. In the second position (dark blue), the first ring faced helix 5 and the other was between helix 6 and 7, forming π-stacking with F6.51 and W6.48. Both poses provided a good fit in the binding pocket, but differed slightly in the conformation of the aryl biphenyl group from the previously published docking results for 1-(2-biphenyl)piperazines. The authors of these studies suggest a second phenyl group facing helix 5. However, these differences may be due to the presence of additional substituents in the biphenyl rings of the mentioned compounds [25]. In the case of **7a**, the ability to form a salt bridge with D3.32 was also observed. However, both the binding donor and acceptor were shifted from **7b**. The compound also showed a different bend of the linker with the terminal heterocycle. The most significant differences appeared in the region of the inner binding pocket, as in **7a** the terminal aryl group moved significantly away from residues F6.51, F6.52 and W6.48, preventing the formation of stable π-stacking. Interestingly, the resulting conformations suggest the importance of the relatively weak π-π interactions for high affinity for the 5-HT_7_R.

### 2.3. ADMET In Silico Evaluation

The preliminary assessment of ADMET properties for compounds **7a** and **7b** was performed using the ADMET Predictor v9.5 [26] platform by analyzing all available parameters (Appendix A). Considering the absorption, the low solubility of **7b** compared to that of **7a** and the reference trazodone should be noted. Due to the basic nature of all the compounds, solubility in the gastric fluids is much more favorable. Based on in silico predictions, compounds **7a** and **7b** were converted to hydrochlorides to improve solubility.

The ability to penetrate the blood–brain barrier was assessed as high for all the ligands. Compounds with a hexyl chain (**7a** and **7b**) may show a lower percentage of the drug unbound to blood plasma proteins. A fairly significant difference between the new compounds and trazodone is seen in the interaction with CYP P450. The likelihood of undesirable interactions was higher for **7a** and **7b** than for trazodone. However, most of them were related to the same isoforms, especially CYP 2C19, CYP 2D6 and CYP3A4. For all compounds analyzed, metabolism was predicted as a clearance mechanism. Characteristics of the OCT2 substrate were not expected.

For **7b**, the properties of the BCRP inhibitor were predicted. Interestingly, a lower risk of mutagenicity was predicted for **7a** and **7b** than for trazodone. All compounds in in silico tests showed no toxic effect and no reproductive toxicity, but they were skin-sensitizing. Despite worse metabolism parameters, **7a** and **7b** showed an ADMET profile comparable to that of trazodone. All parameters were determined with in silico methods and require experimental verification.

### 2.4. Determination of Solubility

Due to the low predicted solubility, we decided to transform compounds **7a** and **7b** into hydrochloride salts (**7a·HCl**; **7b·HCl**) and then experimentally determine their solubility by performing a shake-flask solubility test and analyzing on a UV spectrophotometer at 254 nm. The hydrochlorides (**7a·HCl**; **7b·HCl**) were obtained by dissolving compounds **7a** and **7b** in acetone and precipitating with a 4M solution of HCl in dioxane.

The test compounds showed a solubility of 0.34 mg/mL for **7a·HCl** and 0.21 mg/mL for **7b·HCl**, which allows their use in biological tests. Therefore, we used the compounds in the form of hydrochlorides to carry out all biological studies.

### 2.5. In Vitro Functional Activity Evaluation

Although the receptor affinity of **7a·HCl** and **7b·HCl** for 5-HT_1A_, 5-HT_2A_, 5-HT_6_, 5-HT_7_ and D_2_ receptors was recently disclosed, we evaluated functionally these compounds at 5-HT_7_R (Appendix A), (Appendix A)and 5-HT_1A_R (Appendix A), (Appendix A), based on an ability of a ligand to inhibit cAMP production induced by agonist 5-CT in HEK293 cells (Table 2). We tested compounds **7a** and **7b** as hydrochlorides (**7a·HCl**, **7b·HCl**, respectively). Both behaved as 5-HT_7_R antagonists, while **7b·HCl** additionally acted as a 5-HT_1A_R antagonist in a cAMP assay.

### 2.6. Assessment of Preliminary In Vitro Safety Properties and Inhibition of CYP3A4

In the next step, we verified the properties of the ligand with the less favorable, predicted ADMET profile (**7b****·HCl**) in in vitro studies.

In antidepressant pharmacotherapy, hepatotoxicity is a major concern for safe use, especially considering hepatic failure. Thus, compound **7b·HCl** was evaluated in a hepatoma HepG2 cell line to exclude its potential to produce hepatic failure (Figure 3A). The tested compound did not exhibit hepatotoxic properties in the concentration range of 1–10 µM after long incubation for 72 h. Strong hepatotoxic effects were observed only at higher concentrations of 50 and 100 μM. Consequently, the ligand could only be tested at sufficiently low concentrations, showing no toxicity up to 10 μM, inclusively, which allowed it to be used in the FST tests. CYP isoforms are often responsible for the metabolism of drugs and influence their activity. We decided to investigate the effect of **7b·HCl** on the activity of the most important CYP3A4 isoform. The potential risk of drug–drug interactions (DDI) was examined by a luminescence-based CYP3A4 P450-Glo™ assay (Promega^®^) (Figure 3B). The tested compound **7b·HCl** exhibited a weak ability to create DDI with CYP3A4 (for concentration of 10 µM, the activity of CYP3A4 was inhibited in 35%, whereas the reference ketoconazole (KE) was inhibited completely CYP3A4 at 1 µM). The conducted in vitro ADMET tests allowed initial recognition of the safety profile of the selected substances and the assessment of the possibility of subjecting them to tests on animals.

### 2.7. Behavioral Evaluation

In the next step, the antidepressant-like activity of **7a·HCl** and **7b·HCl** was investigated in the forced swimming test (FST) in mice. Compound **7a·HCl** (0.312–1.5 mg/kg) revealed an antidepressant-like effect in the FST expressed as a reduction of the immobility time of mice by about 23% only at one dose of 0.625 mg/kg, while **7b·HCl** (0.625–2.5 mg/kg) was active at doses of 1.25 and 2.5 mg/kg, decreasing the immobility time by about 40% and 33%, respectively. Thus, the anti-immobility effect of **7b·HCl** was 1.5–2-fold stronger than that of **7a·HCl** and comparable to the effect of escitalopram administered at the same doses (Figure 4).

Moreover, **7a·HCl** given at active dose visibly inhibited spontaneous activity of animals. The results, however, did not reach the level of statistical significance. In contrast, the effect produced by compound **7b·HCl** appeared to be specific because the compound administered at doses evoking antidepressant-like activity did not increase the spontaneous locomotor activity of mice (Table 3).

The conducted in vivo behavioral studies indicate that both tested hexyl analogues of trazodone, **7a·HCl** and **7b·HCl**, having high affinity for the 5-HT_1A_R, showed antidepressant activity. Importantly, in the case of **7b·HCl**, additionally having high affinity for 5-HT_7_R (dual 5-HT_1A_/5-HT_7_ effect), the observed antidepressant effect was much stronger. The anti-immobility effect of **7b·HCl** was even twofold stronger than that of **7a·HCl** and comparable to the effect of escitalopram administered in the same doses. It should be noted that, in contrast to that of **7a·HCl**, the effect induced by compound **7b·HCl** appeared to be specific as the compound administered in the doses inducing antidepressant effects did not increase the spontaneous locomotor activity of mice. The obtained results may indicate a synergistic antidepressant effect of dual 5-HT_1A_/5-HT_7_ receptor antagonists, superior to the activity of selective 5-HT_1A_R ligands. Of course, further behavioral studies and neurological imaging are necessary to confirm this effect.

## 3. Materials and Methods

### 3.1. Chemistry

Chemical reagents were purchased from Sigma Aldrich (Steinheim, Germany). The microwave synthesis was performed in a CEM Discover reactor. Chromatographic analysis (TLC) was performed using Sigma Aldrich sheets (aluminum plates with a silica gel layer of 200 µm thickness and 60 Å pores, with a fluorescence indicator of 254 nm). HPLC analyzes were performed on a Perkin Elmer Series 200 HPLC instrument using a C-18, 3.5 µm, 4.6 × 150 mm column. Melting points were determined with a Boetius apparatus. The synthesis and structural characteristics of compounds for biological research were presented in a previous publication [18]. During the development of the synthesis method, the analysis of the purity of the obtained products and the progress of the reaction were investigated using the HPLC method, using the previously characterized products **7a** and **7b** as a comparative standard. 6-chloro-1,1-dimethoxyhexane (**3**) was synthesized according to a procedure described earlier [29]. NMR analyses were performed on a Bruker Avance 300 MHz spectrometer. UPLC-MS were performed on a Waters Acquity UPLC instrument, coupled to a Waters TQD mass spectrometer, on an Acquity UPLC BEH C-18, 1.7, 2.1 × 100 mm column. The tests were carried out in the electrospray ionization mode in an ESI-tandem quadrupole system. Elemental analysis was performed using Vario EL II.

#### 3.1.1. Synthesis of 6-(3-Oxo[1,2,4]triazolo[4,3-*a*]pyridin-2(3*H*)-yl)hexanal (**5**)

The reaction components: 0.135 g (1 mmol) of [1,2,4]triazolo[4,3-*a*] pyridin-3(2*H*)-one (**1**), 0.414 g (3 mmol) of K_2_CO_3_ and 0.032 g (0.1 mol) of TBAB were weighed. The flowable ingredients were then ground in a mortar, and the mixture was placed in a round bottom flask. Then the contents were suspended in 0.2 cm^3^ of acetonitrile, and 0.18 g (1 mmol) of 6-chloro-1,1-dimethoxyhexane (**3**) was added. The reactions were carried out for 30 s in the presence of 100 W microwave radiation. After the end of the synthesis, 30 cm^3^ of water was added and extracted with three 15 cm^3^ portions of CH_2_Cl_2_. Then, 10 mL of 10% HCl solution was added to the organic layer, followed by stirring for 2 h. After this time, the organic layer was separated, and the solvent was evaporated.

^1^H NMR (400 MHz, CDCl3) δ 7.78–7.72 (m, 1H), 7.16–7.06 (m, 2H), 6.55–6.45 (m, 1H), 4.02–3.89 (m, 2H), 3.44–3.37 (m, 2H), 1.88–1.53 (m, 8 H), t_R_ = 5.91, P = 89%, Y = 89%.

#### 3.1.2. Synthesis of 2-(6-(4-(Aryl)piperazin-1-yl)hexyl)-[1,2,4]triazolo[4,3-*a*]pyridin-3(2*H*)-ones hydrochlorides **7a·HCl** and **7b·HCl** in One-Step Procedures

A mixture of 0.298 g (0.001 mol) of 6-(3-oxo[1,2,4]triazolo[4,3-*a*]pyridin-2(3*H*)-yl)hexanal (**5**), 0.233 g (0.001 mol) of the arylpiperazine (**6a**), 1 cm^3^ of ethanol and 0.08 g (0.002 mol) of NaBH_4_ was transferred to a round bottom flask. The reactions were carried out for 4 min in a CEM Discover microwave reactor at 100 W output power. After this time, 30 cm^3^ of water was added. The crude product after filtration was purified by crystallization from methanol. The resulting compound was converted to a hydrochloride salt by using 4M HCl in dioxane. The identity of the product was confirmed by comparing the retention time and melting point with that of the standard.

##### 2-(6-(4-(3-chlorophenyl)piperazin-1-yl)hexyl)-[1,2,4]triazolo[4,3-*a*]pyridin-3(2*H*)-one hydrochloride **7a·HCl**

t_R_ = 4.32, P = 90%, Y = 16% m_p_ = 175–180 °C.

#### 3.1.3. Synthesis of 2-(6-(4-(Aryl)piperazin-1-yl)hexyl)-[1,2,4]triazolo[4,3-*a*]pyridin-3(2*H*)-ones hydrochlorides **7a·HCl** and **7b·HCl** in Two-Step Procedures

A mixture of 0.298 g (0.001 mol) of 6-(3-oxo[1,2,4]triazolo[4,3-*a*]pyridin-2(3*H*)-yl)hexanal (**5**), 0.001 mol of the corresponding arylpiperazine (**6a**/**b**), was transferred to a round bottom flask and 1 cm^3^ of ethanol was added (or without ethanol in solvent-free conditions). The reactions were carried out for 60 s in a CEM Discover microwave reactor at 100 W output power. After the completion of the reaction, 0.08 g (0.002 mol) of NaBH_4_ was added (in a catalytic reaction, also 0.056g (0.0001 mol) of iron (II) phthalocyanine), and the reaction was continued for 3 min at 100 W output power. After cooling, 30 cm^3^ of water was added, and the crude product was filtered off and crystallized from methanol. The resulting compound was converted to a hydrochloride salt by using 4M HCl in dioxane. The identity of the products was confirmed by comparing the retention time and melting point with that of the standard.

##### 2-(6-(4-(3-chlorophenyl)piperazin-1-yl)hexyl)-[1,2,4]triazolo[4,3-*a*]pyridin-3(2*H*)-one hydrochloride **7a·HCl**

^1^H NMR (300 MHz, CDCl_3_) (Appendix A) δ 7.76 (d, J = 7.1 Hz, 1H), 7.61 (s, 1H), 7.49 (d, J = 8.2 Hz, 1H), 7.39 (d, J = 8.2 Hz, 1H), 7.31 (d, J = 8.3 Hz, 1H), 7.14–7.04 (m, 2H), 6.51 (dt, J = 7.3, 3.7 Hz, 1H), 4.45 (d, J = 11.4 Hz, 2H), 3.91–3.63 (m, 8H), 3.11 (s, 2H), 1.99–1.81 (m, 4H), 1.46 (s, 4H). ^13^C NMR (101 MHz, CDCl_3_) (Appendix A) δ 148.62, 146.74, 141.56, 135.80, 131.06, 129.92, 125.69, 123.72, 119.31, 116.98, 115.375, 110.65, 57.28, 50.21, 48.35, 45.33, 28.32, 26.06, 25.67, 23.39. Formula weight for C_22_H_29_Cl_2_N_5_O: 450.40 g/mol, UPLC-MS (Appendix A): [M + H]^+^ = 414.3, purity = 96%, Y = 56% m_p_ = 177–179 °C. Anal. Calcd for C_22_H_29_Cl_2_N_5_O: C, 58.67; H, 6.49; N, 15.55 Found: C, 58.12; H, 6.43; N, 15.19.

##### 2-(6-(4-(2-phenylphenyl)piperazin-1-yl)hexyl)-[1,2,4]triazolo[4,3-*a*]pyridin-3(2*H*)-one hydrochloride **7b·HCl**

^1^H NMR (300 MHz, CDCl_3_) (Appendix A) δ 7.76 (d, *J* = 7.2 Hz, 1H), 7.52 (d, *J* = 7.2 Hz, 2H), 7.42 (t, *J* = 7.5 Hz, 2H), 7.36–7.29 (m, 2H), 7.24 (s, 1H), 7.21–7.12 (m, 2H), 7.09 (d, *J* = 3.2 Hz, 2H), 6.53–6.47 (m, 1H), 3.99 (t, *J* = 7.0 Hz, 2H), 3.51 (d, *J* = 12.0 Hz, 2H), 3.35 (d, *J* = 10.9 Hz, 2H), 3.10 (d, *J* = 13.4 Hz, 2H), 2.86 (s, 2H), 2.69 (s, 2H), 1.77 (s, 4H), 1.39 (s, 4H), ^13^C NMR (101 MHz, DMSO) (Appendix A) δ 148.71, 148.37, 141.34, 140.59, 131.74, 130.84, 129.03, 128.73, 127.55, 124.29, 119.01, 115.52, 115.44, 111.30, 55.75, 51.25, 47.76, 45.25, 28.42, 26.01, 25.84, 23.25. Formula weight for C_28_H_34_ClN_5_O: 492.06 g/mol, UPLC-MS (Appendix A): [M + H]^+^ = 456.4 purity = 95%, Y = 63%. m_p_ = semi-oil °C. Anal. Calcd for C_28_H_34_ClN_5_O: C, 68.35; H, 6.96; N, 14.23 Found: C, 68.15; H, 6.89; N, 14.42.

### 3.2. Molecular Modelling

#### 3.2.1. Protein-Ligand Docking

The crystal structure of the receptor 5-HT_1A_R in a complex with aripiprazole (pdb id: 7E2Z) was used for 5-HT_1A_ docking. The homology model of the 5-HT_7_R in an inactive state (template 5-HT_1B_R; id: 4IAQ) was obtained from the GPCRdb [23,24]. Ligand models were made using LigPrep 3.7 [30]. The ionization state at pH = 7.4 was determined in the Epik program [31]. Proteins were prepared in Protein Preparation Wizard [32]. Docking was performed in the Schrödinger package [33], using the induced fit (IFD) method. Extended Sampling Protocol was chosen, generating 50 poses for each ligand. During the validation of selected conformations, only those with a coherent binding mode were kept [34].

#### 3.2.2. QM/MM

The obtained conformations were optimized using the hybrid approach (QM/MM) in the quantum polarized ligand docking (QPLD) method in the Schrödinger package [35,36,37,38]. QM calculations were performed on the 3-21G [38], BLYPL theory level. The number of poses returned to the ligand at each docking step was set to 50.

### 3.3. Solubility Tests

The excess 50 mg of **7a·HCl** or **7b·HCl** was added to 100 mL of water. The mixture was stirred at 20 rpm in 20 °C for 48 h. Then, the sample was filtered on a syringe filter (PTFE; 0.20 µm) and analyzed on a UV spectrophotometer at 254 nm. The quantification was performed using the calibration curve method, in the concentration range of 0.05–0.4 mg/mL.

### 3.4. Functional Assays

Functional tests were performed using a LANCE Ultra cAMP test (PerkinElmer). Stimulation of the 5-HT_1A_R (in HEK293 cells) with 1 μM forskolin (EC_90_) reduces the production of cAMP. In the case of 5-HT_7_R, the functional properties of ligands were evaluated using their ability to inhibit cAMP production induced by 5-CT (10 nM), a 5-HT_7b_R agonist, in HEK-293 cells overexpressing 5-HT_7b_R. Cells (prepared with the use of Lipofectamine 2000) were maintained at 37 °C in a humidified atmosphere with 5% CO_2_ and were grown in Dulbeco’s Modifier Eagle Medium containing 10% dialyzed foetal bovine serum and 500 μg/mL G418 sulphate. For functional experiments, the cells were subcultured in 25 cm flasks, grown to 90% confluence, washed twice with prewarmed to 37 °C phosphate buffered saline (PBS) and centrifuged for 5 min (160× *g*). The supernatant was aspirated, and the cell pellet was resuspended in stimulation buffer (1 × HBSS, 5 mM HEPES, 0.5 mM IBMX, 0.1% BSA).

Determination of the change in the amount of cAMP was performed by incubating the cell (5 µL) with a mixture consisting of the test ligand, forskolin and 1 µM of (R)-(+)-8-OH-DPAT. The incubation time was 30 min at room temperature (in a 384-well PerkinElmer microtiter plate). The reaction was then stopped, and the cells were lysed (using 10 µL of a working solution consisting of 5 µL Eu-cAMP and 5 µL ULight-anti-cAMP). Fluorescence resonance energy transfer signal over time was measured on an Infinite M1000 Pro instrument (Tecan). The constants *K*_b_ were determined from the Cheng–Prusoff equation [39]. Each compound was tested in triplicate at 8 concentrations (10^−11^–10^−4^ M).

### 3.5. Safety Tests

The safety parameters of **7b·HCl** were analyzed according to previously described protocols [40,41] and included the influence on CYP3A4 activity and a hepatotoxicity assessment with use of HepG2 cells.

#### 3.5.1. Hepatotoxicity

Liver safety was assessed using the HepG2 cell line (ATCC^®^ HB-8065 ™). The culture was incubated for 72 h in 96-well plates. The compound **7b·HCl** was tested in the concentration range of 1–100 µM. Cell viability was determined using a CellTiter 96^®^ AQueous Non-Radioactive Cell Proliferation Assay (Promega, Madison, WI, USA). Absorbance at 490 nm was determined on an EnSpire reader (PerkinElmer, Waltham, MA USA). One experiment was carried out in quadruplicate using doxorubicin (DX) as the reference drug.

#### 3.5.2. Drug–Drug Interactions

A CYP3A4 P450-Glo™ luminescence assay (Promega, Madison, WI, USA, ketoconazole as reference inhibitor (KE)) was used to evaluate the inhibitory effect of the test compound on CYP3A4. The experiment was performed in triplicate (10 μM). An EnSpire PerkinElmer instrument (Waltham, MA, USA) was used to measure luminescence.

#### 3.5.3. Statistical Analysis

Statistical significance was assessed using GraphPad Prism 8.0.1, in ANOVA followed by Bonferroni’s comparative test (**** *p* <0.0001).

### 3.6. Behavioral evaluation

#### 3.6.1. Animals

The experiments were performed on male Albino Swiss mice (22–28 g), purchased from accredited Laboratory Animal Breeding Ilkowice, Słaboszów, Poland. The animals were kept in an environmentally controlled room (temperature of 22 ± 2 °C, humidity 55 ± 10%) on 12 h light/dark cycles (the lights were turned on at 7:00 a.m. and turned off at 7:00 p.m.) and had free access to food (standard laboratory pellets) and tap water. All the experiments were conducted in the light phase between 9:00 a.m. and 2:00 p.m. Each experimental group consisted of 7–9 animals/dose. The animals were used only once. The experiments were performed by an observer unaware of the administered treatment. All experimental procedures were approved by the I Local Ethics Commission for Animal Experiments of Jagiellonian University in Cracow (no. 41/2018, 1st February 2018).

#### 3.6.2. Drugs

The tested compounds (**7a·HCl,**
**7b·HCl**) were obtained by a previously developed method [18], suspended in a 1% aqueous solution of Tween 80 (Sigma-Aldrich, UK) and injected intraperitoneally (i.p.) 60 min before tests. Citalopram (hydrochloride Adamed Pharmaceuticals, Pieńków, Poland) was dissolved in distilled water and administered i.p. 30 min before the test. All compounds were injected at a volume of 10 mL/kg. Control animals received a vehicle injection according to the same schedule.

#### 3.6.3. Forced Swim Test in Swiss Albino Mice

The experiment was carried out according to the method of Porsolt et al. [42]. Swiss albino mice were individually placed in a glass cylinder (25 cm high; 10 cm in diameter) containing 10 cm of water maintained at 23–25 °C and were left there for 6 min. The total duration of immobility was recorded during the last 4 min of a 6 min test session. A mouse was regarded as immobile when it remained floating on the water, making only small movements to keep its head above it. The shortening of immobility time in comparison to vehicle-treated animals was regarded as antidepressant-like activity.

#### 3.6.4. Locomotor Activity in Mice

The locomotor activity was recorded with an Opto M3 multi-channel activity monitor (MultiDevice Software v.1.3, Columbus Instruments, Columbus, OH, USA). The CD-1 mice were individually placed in plastic cages (22 × 12 × 13 cm) for a 30 min habituation period, and then ambulation was counted from 2 to 6 min, i.e., the time equal to the observation period in the forced swim test. The cages were cleaned up with 70% ethanol after each mouse.

#### 3.6.5. Statistics

All the data are presented as the mean ± SEM. The statistical significance of the results was evaluated by one-way ANOVA, followed by Bonferroni’s Comparison Test, *p* < 0.05 was considered significant.

## 4. Conclusions

We have developed a new, ecological method for the synthesis of LCAP by microwave-assisted reductive alkylation with the use of a mild reducing agent. This method made it possible to obtain **7a** and **7b** with high efficiency (56–63% using ethanol or 51–56% in solvent-free conditions) in four minutes of reaction time. As an alternative to the previously proposed classic microwave-assisted *N*-alkylation reaction, the new method eliminates the need to use a high excess of the alkylating agent. Of note, this method increased the selectivity of the entire process by decreasing creation of a di-substituted intermediate.

Furthermore, the behavioral studies confirmed antidepressant-like activity of *N*-hexyl trazodone derivatives, with a much stronger and more specific effect for compound **7b·HCl**, a dual-acting 5-HT_1A_/5-HT_7_ antagonist, than for a selective 5-HT_1A_R ligand **7a·HCl**. The anti-immobility effect of **7b·HCl** was comparable to the effect of escitalopram administered in the same doses.

## Data Availability

The data presented in this study are available in Appendix A.

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
