# Peer review of "New, Eco-Friendly Method for Synthesis of 3-Chlorophenyl and 1,1′-Biphenyl Piperazinylhexyl Trazodone Analogues with Dual 5-HT1A/5-HT7 Affinity and Its Antidepressant-like Activity"

_molecules, 2022, doi:10.3390/molecules27217270_

Round 1

Reviewer 1 Report

The manuscript entitled "New, eco-friendly method for synthesis of 3-chlorophenyl and 1,1'-biphenyl piperazinylhexyl trazodone analogues with dual 5-HT1A/5-HT7 affinity and its antidepressant-like activity ". In this article, the author synthesized compounds 7a and 7b by another synthetic method, Experiment used in vitro, in vivo and in silico studies. The experimental content is basically comprehensive and the activity is remarkable. This article can be published in Molecules,But there are still many small problems in the article, such as formatting, etc., the author should carefully check and make corrections.

Author Response

The manuscript entitled "New, eco-friendly method for synthesis of 3-chlorophenyl and 1,1'-biphenyl piperazinylhexyl trazodone analogues with dual 5-HT1A/5-HT7 affinity and its antidepressant-like activity ". In this article, the author synthesized compounds 7a and 7b by another synthetic method, Experiment used in vitro, in vivo and in silico studies. The experimental content is basically comprehensive and the activity is remarkable. This article can be published in Molecules,But there are still many small problems in the article, such as formatting, etc., the author should carefully check and make corrections.

  • We thank reviewer for positive opinion and comments. We re-checked the entire manuscript, both in terms of editing and language.

Reviewer 2 Report

The manuscript contains enough significant original material, clearly and concisely written with conclusions adequately supported by data.

In my opinion, this manuscript should be published after minor revision without additional review.

Parts to pay attention to and to be corrected are:

- In lines: 415, 426, 433, 441, and 447, delete the period after the title.

For the References section, the general remark is:

- In the text, reference numbers should be placed in square brackets [ ] and placed before the punctuation;

- References should be described by the requirements in the instructions for authors:

Journal Articles:

1. Author 1, A.B.; Author 2, C.D. Title of the article. Abbreviated Journal Name Year, Volume, page range.

In addition to these general remarks that should be applied to the presentation of all references, the following mistakes should also be corrected:

- the presentation of reference [3], which is given in greyish colour,

- reference [4], which is underlined and presented as [45],

- reference [5], which is presented as [56], and

- reference [6], which is presented as [64].

Author Response

The manuscript contains enough significant original material, clearly and concisely written with conclusions adequately supported by data.In my opinion, this manuscript should be published after minor revision without additional review.

Parts to pay attention to and to be corrected are:

- In lines: 415, 426, 433, 441, and 447, delete the period after the title.

For the References section, the general remark is:

- In the text, reference numbers should be placed in square brackets [ ] and placed before the punctuation;

- References should be described by the requirements in the instructions for authors:

Journal Articles:

  1. Author 1, A.B.; Author 2, C.D. Title of the article. Abbreviated Journal Name Year, Volume, page range.

In addition to these general remarks that should be applied to the presentation of all references, the following mistakes should also be corrected:

- the presentation of reference [3], which is given in greyish colour,

- reference [4], which is underlined and presented as [45],

- reference [5], which is presented as [56], and

- reference [6], which is presented as [64].

  • We thank reviewer for valuable insights. We have made all the recommended corrections. Additionally, we carefully reviewed the entire manuscript for other minor errors.

Reviewer 3 Report

This manuscript reports optimization processes regarding a faster and greener synthesis of two serotoninergic ligands already published with their preliminary ADMET profile determination and in vivo studies. After the chemical synthesis optimization, the manuscript becomes difficult to follow and improvements should be done for reason of clarity. Furthermore, the discussion in every section is poor and should be improved.

In molecular modelling section a comparison should be done among 7a and 7b justifying their different 5-HT7R potency.

Safety profile should be tested before going in vivo, not after. Even if this workflow was not followed during the experimental design (it seems from the manuscript and if it was so it remains ethically not acceptable), at least here it should be presented with the correct and appropriate workflow. Please move the preliminary safety properties evaluation after the ADMET in silico determination. Furthermore, in the ADMET prediction several problems arose that should be experimentally faced before going in vivo. For example, the very low predicted solubility should be experimentally measured and add a comment that this remains feasible with the animal administration. The same should be done experimentally with hERG blocking and proved that this property do not cause its toxicity paired with the cytotoxicity evaluation.

Explain acronyms at first use (i.e., LCAP)

The number 4 beside brackets in scheme should be smaller because it causes confusion with compound numbers.

Compound 7a and 7b are administered to animals, therefore purity assessment should be proved by inserting NMR or HPLC spectra in SI

Author Response

  • This manuscript reports optimization processes regarding a faster and greener synthesis of two serotoninergic ligands already published with their preliminary ADMET profile determination and in vivo studies. After the chemical synthesis optimization, the manuscript becomes difficult to follow and improvements should be done for reason of clarity. Furthermore, the discussion in every section is poor and should be improved.
  • We thank reviewer for comments. We improved the "narratives" of the text and enriched the discussions as recommended.
  • In molecular modelling section a comparison should be done among 7a and 7b justifying their different 5-HT7R potency.
  • We changed Figure 2, supplementing the 5-HT7R molecular modeling sections with compound 7a, as recommended.
  • Safety profile should be tested before going in vivo, not after. Even if this workflow was not followed during the experimental design (it seems from the manuscript and if it was so it remains ethically not acceptable), at least here it should be presented with the correct and appropriate workflow. Please move the preliminary safety properties evaluation after the ADMET in silico determination.
  • We agree with reviewer that ADMET in silico and in vitro studies were performed prior to behavioral studies. We have changed these sections according to the actual order. The wrong order is due to our oversight. We rearranged the layout of the entire manuscript so that the order of the sections corresponded to the order of the tests performed.
  • Furthermore, in the ADMET prediction several problems arose that should be experimentally faced before going in vivo. For example, the very low predicted solubility should be experimentally measured and add a comment that this remains feasible with the animal administration.
  • We agree with reviewer that assessment of ADMET parameters using in silico methods are only predictive. Regarding the recent suggestions to reduce content of in silico prediction only to those in vitro validated data, we moved Table 3 to the Supporting Section. We also thank reviewer for the comment on compound solubility. Indeed, compound 7a and 7b was tested in biological and in vivo pharmacological experiments as hydrochloride salt. We took this information by default. Thank you for paying attention to this fact. We have specified in the text which studies are for free bases (7a and 7b; mainly in silico studies and synthesis) and which are for hydrochlorides (7a·HCl; 7b·HCl; all biological studies). Of note, solubility of compound 7a·HCl and 7b·HCl was verified in distilled water (used as formulation) and it equals to 34 mg/ml for 7a·HCl and 0.21 mg/ml for 7b·HCl.
  • The same should be done experimentally with hERG blocking and proved that this property do not cause its toxicity paired with the cytotoxicity evaluation.
  • The in silico studies were only to indicate which ligands from the library we received (presented in JaÅ›kowska, J.; ZarÄ™ba, P.; et al. Molecules 2019, 24, 1609–1628) should be selected for further research, taking into account the ADMET profile. In Table 3, we included only the prediction results for the most favorable compounds (7a and 7b) and compared them with a well-tested compound with a similar structure - trazodone. Table 3 does cause chaos in the manuscript content, so we've moved it to the supplementary material. As for the expected blocking of hERG - we obtained similar indications in the predictive model for trazodone, an antidepressant drug known for many years, with a structure very similar to the tested compounds 7a and 7b. Due to the very similar structural properties and predicted affinity for hERG, at this stage we did not consider this parameter as critical for the rejection of candidates for in vivo We focused on the in vitro hepatotoxicity assessment, which we did not assess in silico and DDI, due to the in silico prediction. The conducted behavioral studies were aimed at showing the benefits of using a dual 5-HT1A/5-HT7 antagonist over a selective 5-HT1A antagonist. Therefore, we concluded that this parameter (predicted ability to inhibit hERG) is not a basis for abandoning behavioral studies. Obviously, a more accurate evaluation of the ADMET profile as well the safety profile and ADME parameters (including hERG test)will be necessary for further research into the use of 7b.
  • Explain acronyms at first use (i.e., LCAP)
  • We reviewed the manuscript for unexplained abbreviations and filled in the gaps.
  • The number 4 beside brackets in scheme should be smaller because it causes confusion with compound numbers.
  • We improved this element in all diagrams.
  • Compound 7a and 7b are administered to animals, therefore purity assessment should be proved by inserting NMR or HPLC spectra in SI
  • We added the missing analyzes to the supporting material.

Round 2

Reviewer 3 Report

The authors have properly answered to reviewer's questions, anyway some minor points remain to be clarified:

1) in reviewer response 7aHCl's solubility is 34 mg/ml while in the manuscript remain 0.34 mg/ml. Please clarified.

2) please modify the TOC accordingly to the new manuscript tale

3) Paragraph 2.3 should be implemented with the most interesting data obtained (referring also to the table now moved in SI, which is not cited) from the computational predictions.

Author Response

  • 1) in reviewer response 7aHCl's solubility is 34 mg/ml while in the manuscript remain 0.34 mg/ml. Please clarified.
  • We thank reviewer for positive opinion and comments. We re-checked the entire manuscript, both in terms of editing and language.We apologize for the error in the text of the answer. The correct solubility value is 0.34 mg / ml for 7a HCl and 0.21 mg / ml for 7b HCl,
  • 2) please modify the TOC accordingly to the new manuscript tale
  • We rearranged the TOC so that the order corresponds to the content of the manuscript.
  • 3) Paragraph 2.3 should be implemented with the most interesting data obtained (referring also to the table now moved in SI, which is not cited) from the computational predictions.
  • In the text, we have added the most interesting data obtained from the computational predictions. We also added a link to the table with ADMET in silico results (Table 1-SI).